# Myostatin and Follistatin—New Kids on the Block in the Diagnosis of Sarcopenia in IBD and Possible Therapeutic Implications

**DOI:** 10.3390/biomedicines9101301

**Published:** 2021-09-23

**Authors:** Dorota Skrzypczak, Marzena Skrzypczak-Zielińska, Alicja Ewa Ratajczak, Aleksandra Szymczak-Tomczak, Piotr Eder, Ryszard Słomski, Agnieszka Dobrowolska, Iwona Krela-Kaźmierczak

**Affiliations:** 1Department of Gastroenterology, Dietetics and Internal Diseases, Poznan University of Medical Sciences, Przybyszewskiego Street 49, 60-355 Poznan, Poland; alicjaewaratajczak@gmail.com (A.E.R.); aleksandra.szymczak@o2.pl (A.S.-T.); piotreder@ump.edu.pl (P.E.); agdob@ump.edu.pl (A.D.); 2Institute of Human Genetics, Polish Academy of Sciences Poznan, Strzeszynska Street 32, 60-479 Poznan, Poland; mskrzypczakzielinska@gmail.com (M.S.-Z.); slomski@up.poznan.pl (R.S.)

**Keywords:** sarcopenia, IBD, myostatin, follistatin

## Abstract

Sarcopenia, which is a decrease in muscle strength and quality of muscle tissue, is a common disorder among patients suffering from inflammatory bowel disease. This particular group of patients often presents with malnutrition and shows low physical activity, which increases the risk of sarcopenia. Another important factor in the development of sarcopenia is an imbalanced ratio of myostatin and follistatin, which may stem from inflammation as well as genetic factors. Currently, research in this area continues, and is aimed at identifying an effective medication for the treatment of this condition. Additionally, we still have no sarcopenia markers that can be used for diagnosis. In this paper, we address the role of myostatin and follistatin as potential markers in the diagnosis of sarcopenia in patients with Crohn’s disease and ulcerative colitis, particularly in view of the genetic and biological aspects. We also present data on new perspectives in the pharmacotherapy of sarcopenia (i.e., myostatin inhibitors and gene therapy). Nevertheless, knowledge is still scarce about the roles of follistatin and myostatin in sarcopenia development among patients suffering from inflammatory bowel disease, which warrants further study.

## 1. Introduction—The Function of Myostatin and Follistatin in IBD Patients Suffering from Sarcopenia

Sarcopenia is characterised by decreased muscle strength and a reduced number and quality of muscle cells [1]. Although in the past, sarcopenia was associated with aging, it may also be a disorder occurring at every age, including childhood [2,3]. Sarcopenia constitutes a consequence of numerous diseases, such as inflammatory bowel disease (IBD) [1,4,5,6], chronic obstructive pulmonary disease [7], hepatic cirrhosis [8], chronic kidney disease [9] and various neurodegenerative diseases (e.g., Duchenne Muscular Dystrophy [10] or Parkinson’s disease) [11].

Malnutrition is a frequently discussed risk factor of sarcopenia among IBD patients, and currently, diet therapy [12,13,14] and physical activity [15] are the most essential elements of sarcopenia treatment.

Medications decreasing or inhibiting the progress of sarcopenia have already been developed [16,17,18]; nevertheless, it is vital to note that methods (e.g., biochemical markers) used for the diagnosis and monitoring of sarcopenia are scarce. Therefore, further research in this area is necessary.

According to the data, patients suffering from IBD are at a higher risk of malnutrition due to malabsorption. Sarcopenia is common among this group [1], and affects approximately 51–70% of patients [12,13,14]. In fact, 40% of IBD patients also suffer from non-alcoholic fatty liver disease (NAFLD), or even its more severe stage—non-alcoholic steatohepatitis (NASH) [19,20].

This article focuses on the genetic and biological mechanisms of myostatin and follistatin.

### 1.1. Myostatin

Myostatin (MSTN, GDF 8—growth differentiation factor 8), a highly conserved member of the transforming growth factor-β superfamily, is a negative regulator of muscle growth and development [21,22]. This protein occurs predominantly in the skeletal muscle tissue, although a decreased amount of myostatin is also observed in the adipose tissue, as well as in the cardiac muscle. Myostatin interacts with the activin type IIB (ActRIIB) receptor. Disorders in the functions of myostatin (e.g., as a consequence of mutation) lead to hyperplasia (an increase in the number of muscle cells) and hypotrophy (an increase in the size of muscle strands), which result in hypermuscular phenotypes [22,23]. Therefore, a modification of the myostatin metabolic pathway may constitute a contemporary way to inhibit the development of sarcopenia, or reverse the effects of the disease. Moreover, it may also protect against cachexia, and support the regeneration of the muscle tissue following trauma or neurodegenerative diseases [22].

It has been established that its serum concentration increases with age, and is inversely proportional to skeletal muscle mass. The metabolism of myostatin is complex, and its functionality is subject to various regulatory mechanisms (activation of the ActRIIB receptor-binding protein by BMP1 (bone morphogenetic protein 1)), and interactions with the three most important myostatin regulatory proteins: GASP (GDF-associated serum protein-1), follistatin, and FLRG (follistatin-related gene) [24]. Additionally, the research study demonstrated a correlation between the serum myostatin concentration and the adipose tissue mass. In fact, in patients with sarcopenic obesity, an elevated myostatin level is correlated with insulin resistance [25]. In contrast, in individuals undergoing diets, myostatin levels in the blood decrease, and the cellular response to insulin is normalised [26].

### 1.2. Follistatin

Follistatin is known as an inhibitor of transforming growth factor (TGF)-β superfamily ligands, including myostatin and activin A. It is a single chain glycoprotein and a specific FSH inhibitor. Follistatin was isolated for the first time from ovarian follicle by Robertson and Ueno in 1987 [27,28]. It is expressed in various tissues and is an antagonist of proteins from TGF-β group, including activin, myostatin and BMP (a bone morphogenetic protein). Additionally, follistatin is also one of the proteins that participates in responses to environmental stress (e.g., oxidative stress, hypoxia, or nutritional deficiency) and constitutes an element of cell responses to stress [29].

The abnormal expression of follistatin may be associated with various liver diseases, including NAFLD/NASH, hepatocellular carcinoma, and fibrosis [30]. In fact, studies show that follistatin can accelerate the progression of NALFD to NASH [31].

Oxidative stress, i.e., the imbalance between creation and elimination of ROS (reactive oxidative species), is the underlying cause of many disorders in the elderly [32]. However, follistatin reduces the number of ROS, whereas a deficiency of follistatin may lead to oxidative stress and result in numerous diseases, including sarcopenia and NAFLD [33,34]. These two diseases also show similarities in pathophysiology (vitamin D deficiency, insulin resistance, inflammation, obesity and too low physical activity) [35]. Nevertheless, with age, the number of ROS in muscles increases and the concentration of myostatin increases [36,37,38], while follistatin concentration is decreased.

## 2. Genetic Factors in Sarcopenia Affecting Patients with IBD

Sarcopenia, which is a common issue in patients with IBD, has been associated with fatigue and a reduction in the quality of life [1]. The cause of skeletal muscle atrophy is complex and, according to the literature, it is comprised of both environmental factors and genetic predispositions, as well as the interaction between both the environment and genetics [39,40]. Moreover, studies regarding heredity have revealed a strong genetic determination for muscle mass and muscle strength, ranging between 32–67% and 46–76%, respectively [39]. The aetiology of IBD is also multifactorial, and includes a substantial hereditary background. According to studies conducted in various centres around the world, it is estimated that in the group of 300 candidate genes, there are currently at least 40 genes that may be responsible for the development of IBD [41,42]. However, the question emerges as to whether the genetic factors of both phenotypes, i.e., muscular and gastroenterological disorders, can be related or interact with each other in IBD patients with sarcopenia. The experimental studies regarding the aforementioned aspect have not been described in the literature so far; nevertheless, it is worth summarizing the current knowledge from this new perspective. Inflammatory processes can be expected to share certain common elements, but it may not be as simple as that (Figure 1) [43,44].

The studies regarding the molecular genetics of sarcopenia gene identification include many different types of research, i.e., whole-genome linkage studies, quantitative trait loci mapping, candidate gene association studies, genome-wide association studies, and DNA microarrays, as well as microRNA studies of sarcopenia, or a related skeletal muscle phenotype. This variety results in a large quantity of data, which are frequently contradictory [39,40,45]. The most frequently mentioned genes correlated with the occurrence of sarcopenia include *ACTN3*, *ACE* and *VDR*, although in general, the list is more extensive, e.g., in the 54 studies of adults aged > 50 years, 26 genes and 88 DNA polymorphisms were analysed [39]. Therefore, in this paper, we focused only on the genetic relationships of sarcopenia, which potentially shares a common element with IBD or with the treatment response summarized in the Appendix A.

From the 30 IBD regions described in the Online Mendelian Inheritance in Man^®^ (OMIM) database, seven regions shared the same localization on chromosome 1p, 7p, 7q, 10q, 12p, 12q and 18p as an association with the muscle-related phenotypes. Further research has contributed to expanding the available knowledge, and associations with both sarcopenia and IBD have been demonstrated for the following receptor genes: vitamin D receptor (*VDR*, MIM:601769) and glucocorticoid receptor (*NR3C1*, nuclear receptor subfamily 3 group C member 1, MIM:138040). The relationship was also found for the following cytokine genes: interleukin 6 (*IL6*, MIM:147620), interleukin 10 (*IL10*, MIM 124092), tumour necrosis factor (*TNF*, MIM:191160), lymphotoxin alpha (*LTA*, MIM:153440) (Appendix A).

*VDR* is one of the most frequently studied candidate genes for sarcopenia, due to its crucial regulatory role in calcium homeostasis and the skeletal muscle function. Thus, the *VDR* gene is a common research object, although the findings are often incompatible [39,40,45]. In general, nine studies with a total of five polymorphisms of this gene, including rs1544410 (BsmI), rs2228570 (FokI), rs7975232 (ApaI) and rs7136534, demonstrated a significant relationship with the sarcopenia phenotype (Appendix A). VDR activates the signal transduction pathways in the skeletal muscle cells, through which vitamin D regulates contractility and myogenesis [46]. Interestingly, the analysis regarding the addition of 1.25(OH)_2_ Vitamin D3 to C2C12 myoblasts clarified the vitamin D-related molecular pathways for muscle regulation. Specifically, 1.25(OH)_2_ Vitamin D3 leads to: (1) a greater expression and nuclear translocation of the vitamin D receptor; (2) a reduced cell proliferation and a decreased IGF-I expression; and (3) the promotion of myogenic differentiation by increasing IGF-II and follistatin expression as well as reducing the expression of myostatin—the only known negative regulator of muscle mass [47]. The *VDR* gene polymorphisms in two studies on IBD susceptibility also indicated a positive correlation (Appendix A). This correlation is substantiated by the local action of 1.25(OH)_2_ Vitamin D3 as a cytokine, following the synthesis and release by the activated macrophages. In addition, 1.25(OH)_2_ Vitamin D3 can inhibit the nuclear factor (NF)-kB and lead to the production of various different cytokines. Studies with animal models demonstrated that the loss of *VDR* expression in VDR-knockout mice resulted in impaired inflammation reactions in the gastrointestinal tract [48].

An association with the muscle-related phenotypes (a greater body height and an increased muscle mass and strength in young-adult males) has also been shown in the variant for the glucocorticoid receptor (GR) gen—*NR3C1*. Moreover, it has been demonstrated that these gene polymorphisms determine steroid therapy outcomes in IBD patients (Appendix A), since glucocorticoids (GCs) are often clinically used as anti-inflammatory and immunosuppressive agents. *NR3C1* is currently characterized as the key gene for the effects of GCs therapy. In fact, its mutations and polymorphisms may influence the formation of GR–GCs complex and change the expression of the target genes. The E22R/E23K (rs6189/rs6190) polymorphisms are located in the N-terminal transactivation domain of the GR, and influence the reduction in sensitivity for GCs [49]. On the other hand, GCs may decrease the rate of protein synthesis and increase protein breakdown, resulting in muscle atrophy. However, in terms of muscle phenotype, the exact mechanism of *NR3C1* polymorphisms in action at the molecular level is unknown. According to the recent in vitro research, human myoblasts and myotubes treated with dexamethasone presented an enhanced mRNA expression in their myogenic proliferation and differentiation markers, although myoblasts were limited in their differentiation potential [50].

The search for common genetic factors for sarcopenia and IBD also revealed *IL6*, *IL10*, and *TNF*, as well as *LTA*, which encodes a multifunctional cytokine primarily involved in the immune functions. In particular, IL-6 has been associated with many inflammatory disorders, presumably due to its robust effects on the induction of pathogenic Th17-cell differentiation. IL-10 plays a critical regulatory role by means of promoting the expression of anti-inflammatory effectors, and is involved in the maintenance of intestinal mucosal homeostasis in the gastrointestinal tract [51]. On the other hand, TNF mediates pleiotropic effects—apoptosis, cell proliferation and differentiation, and cytokine production through binding to TNFR-I and TNFR-II receptors in the targeted tissue. LTA (also known as TNF-β) is produced by lymphocytes and participates in numerous immunological processes, and may also be associated with anti-TNF responses [52]. However, IBD research results regarding the aforementioned cytokine genes are also contradictory. The Appendix A presents only the genes with a positive correlation. Additionally, chronic inflammation has been also considered to be involved in the development of sarcopenia, due to the production of numerous pro-inflammatory cytokines, such as TNF and IL-6, which promote muscle catabolism [53]. Moreover, IL-6 signalling affects the binding to the membrane-bound IL-6 receptor in the skeletal muscle, as well as the activation of downstream signalling pathways including MAPK/ERK, STAT3, p38, FoxO3 and myostatin pathways [54,55].

In recent years, numerous research studies have been performed and considerable attention has been devoted to the microRNA (miRNA) molecules. In fact, these molecules have been regarded as potent regulators in diverse biological processes at the post-transcriptional level. Moreover, proliferation and differentiation of the muscle stem cells requires the regulations signals for the optimal regeneration and the functioning of the muscles [56]. MiRNAs involved in muscle biology and the pathophysiology of IBD are presented in the Appendix A—they were selected on the basis of the available lists for both diseases [40,56,57]. We found six molecules to have their expression regulated in both sarcopenic and IBD patients: mi-206, mi-146a, mi-19, mi-21, mi-27, and mi-201. Nonetheless, the Appendix A only includes the first two, since the available information for the remaining molecules regarding the function, signalling pathways, targets, and regulators is unknown or limited, particularly in the case of IBD. However, it is worth emphasizing that regulatory miRNAs, or other non-coding molecules, may be the key to explaining the coexistence of sarcopenia and IBD.

Overall, although there are currently indications to look for the genetic factors determining sarcopenia in patients with IBD, no strong genetic evidence between the two exists. Nevertheless, many ambiguities may be addressed soon, owing to non-coding RNA research. This may provide knowledge of new pathways in the mechanisms of the abovementioned diseases, as well as diagnostics and therapy.

### 2.1. Myostatin (MSTN) Gene

The myostatin gene (*MSTN,* also referred to as the growth differentiation factor 8, *GDF8*; MIM601788), encodes a skeletal muscle-specific secreted preproprotein, which is proteolytically processed to generate each subunit of the disulphide-linked homodimer [58]. Moreover, myostatin is a highly conserved member of the transforming growth factor-β superfamily, which is expressed mostly in the muscle tissue, and negatively regulates skeletal muscle cell proliferation and differentiation. Thus, mutations in *MSTN* gene are associated with an increased skeletal muscle mass in humans and other mammals [59]. One example is splicing variant c.373+5G>A (rs397515373), which causes large muscle hypertrophy and unusual strength in children with the homozygous variant [23].

According to the human genome reference assembly GRCh38.p13, this relatively small gene is located on chromosome 2 (2q32.2), between the coordinates 190,055,699 and 190,062,728 bp (Figure 2). It comprises 7029 bp of genomic DNA, and includes only three coding exons. The transcript length is 2819 bp, and the translation product consists of 375 amino acids containing five missense substitutions in conserved amino acid residues: p.A55T (rs1805085, c.163G>A), p.K153R (rs1805086, c.458A>G), p.E164K (rs35781413, c.490G>A), p.P198A (rs368949692, c.592C>G), and p.I225T (rs143242500, c.674T>C) (Figure 2). Changes in p.A55T in exon 1 and p.K153R in exon 2 are polymorphic in the general population, and they present significantly different allele frequencies in Europeans and African Americans (0% and 3% vs. 16% and 22%, respectively, *p* < 0.001). Simultaneously, the change in p.K153R (rs1805086, c.458A>G) in the heterozygous status was described as influencing the clinical presentation of women with McArdle disease [60].

### 2.2. Follistatin (FST) Gene

The follistatin gene (*FST,* MIM136470), encoding a highly conserved among the species cysteine-rich monomeric glycoprotein, is located on chromosome 5 (5q11.2), contains 6797 bp between coordinates 53,480,338 and 53,487,134 bp (GRCh38.p13), and comprises six exons (Figure 3). The follistatin, similarly to myostatin, is included in the TGF-beta superfamily. Moreover, the product of the *FST* gene is synthesized as pre-follistatin in isoforms 344 and 317 amino acids depending on the transcript, which differs in length at the C-terminus [61]. The mature protein was formed following the cleavage of the signal peptide (29 amino acids) (Figure 3). Different genetic variants of the *FST* gene have been described in the literature, such as the coding region, e.g., rs11745088 (c.454G>C, p.E152Q) and rs1127760 (c.715T>A, p.C239S). However, they were analysed in the context of susceptibility to polycystic ovarian syndrome (POS)) [62,63], or premature ovarian failure (POF) [64]. This line of research results from linkage and association studies [65], as well as from animal studies, which suggest that the FST loss may result in a premature ovarian cessation [66]. Further results are controversial, since some support this thesis [63], whereas other studies found no evidence that *FST* is a POS or POF disease-causing gene [62,64]. In terms of sarcopenia, the intronic variants rs3797297 (c.85+950G>T), rs3756498 (c.86−500G>A), rs12152850 (c.721+230C>T), and rs12153205 (c.722−241T>C) in the *FST* gene have all been investigated to date, and comprise a haplotype block [67]. The obtained data have demonstrated that *FST loci* may contribute to the interindividual variation in both skeletal muscle mass and strength in men, but not in women [67]. However, similar studies of *FST* gene variants have not been conducted so far.

## 3. The Pharmacotherapy of Sarcopenia—New Perspectives

### 3.1. Inhibitors of Myostatin and Other Myokines

The discovery of myostatin and its function has initiated a new phase of research into the treatment of sarcopenia. In mice models, the pharmacological blockade of myostatin induces muscle mass gain and improves metabolic management of insulin, triglycerides, and circulating adiponectin in obese mice with insulin resistance (128). Mice with either mutated or defective myostatin have been shown to be resistant to diet-induced obesity, dyslipidaemia, atherogenesis, and hepatic steatosis. Researchers are primarily interested in the antibodies against myostatin, propeptides, soluble ActRIIB receptors, and interacting proteins (GASP-1, follistatin and FLRG) [22,68,69,70].

The first myostatin inhibitor investigated in humans was Stamulumab (MYO-029) developed by Wyeth Pharmaceuticals of the Cambridge Antibody Technology Group. It is a recombinant human antibody that blocks the binding of myostatin to ACVR2B. Phase 2 clinical trials were conducted in patients with muscular dystrophy; however, these were discontinued due to a lack of efficacy in improving muscle strength [71]. Landogrozumab (LY2495655) (Eli Lilly and co) is a humanised κ-type IgG4 monoclonal antibody that binds to and neutralises the MSTN protein, which blocks the MSTN signalling pathway. This process leads to reduced protein breakdown and decreased muscle weakness. The study by Becker et al. of 365 elderly patients with a history of falls found an increase in the total lean body mass and a decrease in the fat mass in the group treated with Landogrozumab vs. placebo. Additionally, the implementation of LY2495655 also led to an overall increase in physical performance in fast gait tests, stair climbing tests, and chair rise tests [72]. Other investigated anti-myostatin antibodies include Trevogrumab and Domagrozumab (PF-06252616) (132). There is also ongoing research on the antibody against activin A (REGN-2477), which together with myostatin transmits a signal to the target cell [73].

Ramatercept (ACE-031) is an antibody that binds ActRIIB ligands, including myostatin, GDF11, and activins, thus blocking the interaction of bound ligands with endogenous ActRIIB receptors. However, the use of this drug was associated with numerous adverse reactions, mainly haemorrhage. Hence, further studies were abandoned. ACE-083, which also binds TBG-β molecules, but does not bind BMP9/10 ligands, is also currently being investigated [74]. ACE-083 was designed for clinical trials in patients with muscular dystrophy.

Bimagrumab (BYM-338) is a monoclonal antibody that blocks the type II receptor for activin and stimulates skeletal muscle growth by enhancing myoblast differentiation, and by counteracting differentiation induced by myostatin or activin A. The antibody acts by inhibiting Smad2/3 phosphorylation, which protects the myosin-heavy chain from degradation. BYM-338 may also counteract glucocorticosteroid-induced muscle atrophy, which could be promising for IBD cases where glucocorticosteroids are used to treat the exacerbations of the disease [75,76].

The effects of the pharmacological administration of follistatin are not limited to the skeletal muscles, since it may affect other tissues and organs by modulating various molecules, activin A, inhibin, follitropin, and certain BMP proteins, and can compromise the pineal gland and gonadal function [22]. Studies investigating the administration of soluble ActRIIB receptors in postmenopausal women and patients with Duchenne Muscular Dystrophy were discontinued due to the occurrence of nose and gum haemorrhage, as well as the appearance of skin lesions [77].

However, it is worth emphasising that the aforementioned drugs have not yet been tested in a specific group of patients with IBD. Thus, further research is necessary to develop the most effective methods of blocking myostatin in order to safely increase the muscle mass and strength in patients with sarcopenia.

### 3.2. Gene Therapies for Sarcopenia

In addition to these therapeutic strategies in the treatment of sarcopenia, gene therapies remain a source of great interest [78]. Interestingly, Morine et al. described a unique method of inhibiting myostatin by using a recombinant adeno-associated virus to overexpress the secreted dominant negative myostatin in the liver of mice. As a result, myostatin inhibition led to an increase in muscle mass and strength in control mice [79].

## 4. Summary and Conclusions

Sarcopenia in IBD is a common but still poorly understood disorder. At present, the treatment is based on proper nutrition and physical exercise, whereas pharmacological treatment is still being investigated. Moreover, there are no biochemical sarcopenia markers. Therefore, the assessment of follistatin and myostatin levels seems necessary, particularly in patients suffering from IBD, who are at an increased risk of developing this condition. In fact, it may constitute the basis for research focused on investigating sarcopenia markers. Moreover, genetic factors may also affect the level of myostatin and follistatin. Hence, the search for the polymorphism responsible for coding these proteins may be essential in order to develop a better understanding of sarcopenia.

## Figures and Tables

**Figure 1 biomedicines-09-01301-f001:**
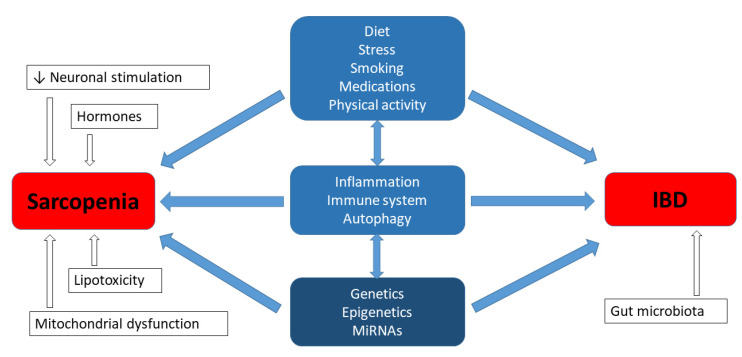
Genetics in aetiology of sarcopenia and IBD.

**Figure 2 biomedicines-09-01301-f002:**
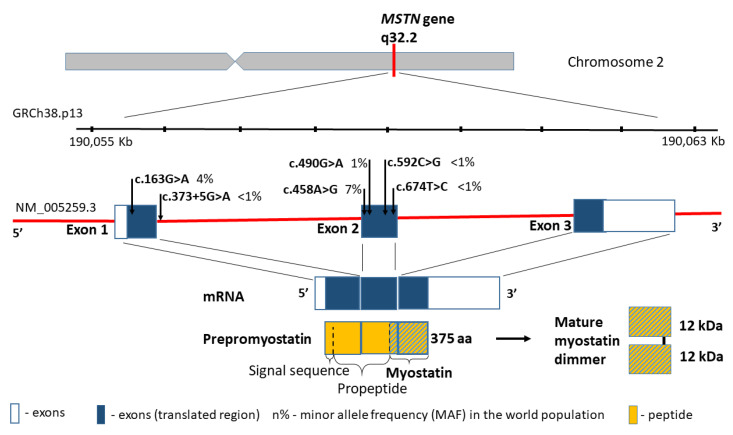
*MSTN* gene structure, variants distribution and the protein product.

**Figure 3 biomedicines-09-01301-f003:**
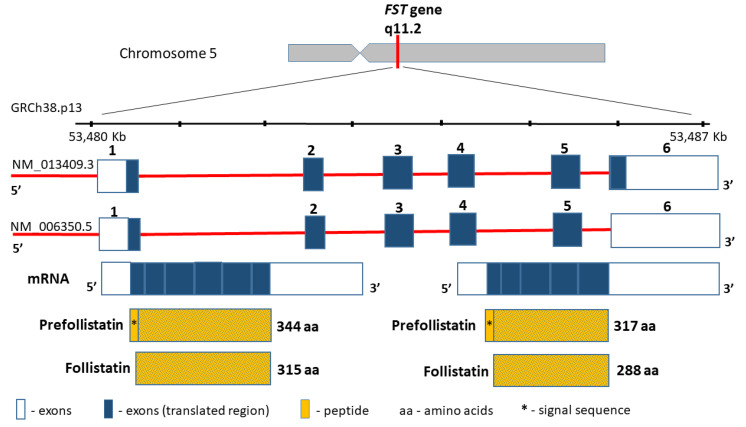
*FST* gene structure and protein product.

## Data Availability

Not applicable.

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
