# Peer review of "Myostatin and Follistatin—New Kids on the Block in the Diagnosis of Sarcopenia in IBD and Possible Therapeutic Implications"

_biomedicines, 2021, doi:10.3390/biomedicines9101301_

Round 1
Reviewer 1 Report
Nonalcoholic fatty liver disease (NAFLD) is the most common chronic liver disease, even though its inner mechanisms and the associated morbidities are far from being clarified, as evident in ....J. Clin. Med. 2020, 9(1), 15.
NAFLD is frequently reported in patients with inflammatory bowel disease (IBD), even though the underlining causes and predisposing factors to NAFLD among IBD patients remain poorly investigated.
Within the NAFLD group of patients, follistatin was associated with NASH independently from activin A, gender and age. This relationship is likely reflecting the effect of adiposity, as evident in.... Activin A and follistatin in patients with nonalcoholic fatty liver disease. Metabolism. 2016;65(10):1550-1558.
Considering the importance of myostatin in NAFLD as a potent growth factor that determines muscle size, as evident in....Targeting myostatin for therapies against muscle-wasting disorders. Curr Opin Drug Discov Devel. 2008 Jul;11(4):487-94 and in.... Role of Oxidative Stress in Hepatic and Extrahepatic Dysfunctions during Nonalcoholic Fatty Liver Disease (NAFLD). Oxid Med Cell Longev. 2020 Oct 19;2020:1617805..... and .....at the light that Individuals with lower muscle mass (sarcoprenic obesity) exhibited increased risk of NAFLD, providing a novel insight into the mechanism linking between sarcopenia and NAFLD, as evident in ...Relationship between sarcopenia and nonalcoholic fatty liver disease: the Korean Sarcopenic Obesity Study. Hepatology. 2014 May;59(5):1772-8, authors should expand this aspect to give readers a broader view of the topic.
Author Response
Dear Reviewer,
Many thanks for all your comments and suggestions. I find them very interesting and inspiring. I`ve read the literature you proposed and even some more and I decided to include it all in my publication. I hope the new shape of the article will find your appreciation.
Kind regards,
Dorota Skrzypczak
Reviewer 2 Report
Paper “Myostatin and follistatin – new kids on the block in the diagnosis of sarcopenia in IBD and possible therapeutic implications?” sheds a new light on the sarcopenia in inflammatory bowel disease and creates a background for the further studies.
In my opinion part of the paper focused on the genetic background of sarcopenia in IBD is too broad compared to the parts on myostatin and follistatin which are the topic of manuscript. Abstract does not fully correspond with the content of the paper.
I have minor remarks:
- English should be improved. There are many speeling mistakes in the manuscript – for instance fig. 1 should be physical not physican etc.
- References should be adjusted to the requirements of the journal – in the text reference numbers should be placed in square brackets etc.
- All abbreviations should be explained when they are first time used (for instance paragraph 1.2 BMP or FST).
- Paragraph 1.1 – sentence “In fact, in patients with obesity, myostatin level correlated with insulin resistance sarcopenic obesity” is unclear.
- Paragraph 1.2 - sentences “However, follistatin reduces the number of ROS, whereas a deficiency of follistatin may lead to oxidative stress, resulting in numerous diseases, including sarcopenia. Research shows that follistatin FST reduces ROS and that a deficiency of this protein can induce oxidative stress and lead to the progression of a variety of diseases, including sarcopenia” are just a repetition.
- Paragraph 2 – first sentence is a repetition from the introduction.
Author Response
Dear Reviewer,
I`d like to thank a lot for all your suggestions. I find them really inspiring and necessary.
I tried to include them all in my publication. I hope that new version will gain your appreciation.
Your faithfully,
Dorota Skrzypczak
Round 2
Reviewer 1 Report
Authors correctly answered comments